# Co-Developing an Antibiotic Stewardship Tool for Dentistry: Shared Decision-Making for Adults with Toothache or Infection

**DOI:** 10.3390/antibiotics10111345

**Published:** 2021-11-04

**Authors:** Wendy Thompson, Jonathan Sandoe, Sue Pavitt, Tanya Walsh, Lucie Byrne-Davis

**Affiliations:** 1Division of Dentistry, University of Manchester, Manchester M13 9PL, UK; tanya.walsh@manchester.ac.uk; 2School of Dentistry, University of Leeds, Leeds LS2 9JT, UK; S.pavitt@leeds.ac.uk; 3School of Medicine, University of Leeds, Leeds LS1 3EX, UK; j.sandoe@leeds.ac.uk; 4Division of Medical Education, University of Manchester, Manchester M13 9PL, UK; lucie.byrne-davis@manchester.ac.uk

**Keywords:** antibiotic, stewardship, decision making, shared, dental, toothache, infection, primary healthcare, behavioural influences, dental procedures

## Abstract

Dentistry is responsible for around 10% of antibiotic prescribing across global healthcare, with up to 80% representing inappropriate use. Facilitating shared decision-making has been shown to optimise antibiotic prescribing (antibiotic stewardship) in primary medical care. Our aim was to co-develop a shared decision-making antibiotic stewardship tool for dentistry. Dentists, patients and other stakeholders prioritised factors to include in the new tool, based on previous research (a systematic review and ethnographic study) about dentists’ decision-making during urgent appointments. Candidate behaviour-change techniques were identified using the Behaviour Change Wheel and selected based on suitability for a shared decision-making approach. A ‘think aloud’ study helped fine-tune the tool design and Crystal Marking ensured clarity of messaging. The resulting paper-based worksheet for use at point-of-care incorporated various behaviour change techniques, such as: ’information about (and salience of) health consequences’, ‘prompts and cues’, ‘restructuring the physical (and social) environment’ and ‘credible sources’. The think aloud study confirmed the tool’s acceptability to dentists and patients, and resulted in the title: ‘Step-by-step guide to fixing your toothache.’ Further testing will be necessary to evaluate its efficacy at safely reducing dental antibiotic prescribing during urgent dental appointments in England and, with translation, to other dental contexts globally.

## 1. Introduction

Antimicrobial resistance is a significant threat to global health, wealth and well-being, and is driven by the use of antibiotics [1]. The World Health Organisation global action plan on tackling antimicrobial resistance, therefore, seeks to optimise the use of antimicrobials in human and animal health (known as antimicrobial stewardship) [1]. The United Kingdom (UK) Government 20-year vision is for strong antimicrobial stewardship and diagnostic stewardship, by ensuring all decisions are supported by diagnostic tests and decision-support tools [2]. Dentistry accounts for around 10% of antibiotic (antibacterial drug) prescribing across international healthcare, with up to 80% shown to represent overprescribing [3]. In England’s publicly funded National Health Service (NHS), urgent dental care for people with acute dental pain or infection accounts for most dental antibiotics, with around 90% of them for adults [4]. In 2020, 3.0 million antibiotics were dispensed to dental patients, costing NHS England £7.6M [5].

Unlike in medical settings, where many common infections are amenable to self-care [6], acute dental pain and infection will usually recur without a procedure [7]. As dental infections have the potential to spread rapidly to become life-threatening conditions, all dental surgeons are skilled to diagnose and manage dental pain and infection using dental procedures (such as extraction of a tooth) usually without the need for antibiotics [3]. In addition, all dental practices in the UK are equipped to diagnose bacterial infections during consultations, for example with radiographs [8]. Safely reducing dental antibiotic prescribing for people with people with acute dental pain or infection must, therefore, be associated with an increase in the number of dental procedures.

Compared to medicine, few antibiotic stewardship interventions have been developed to optimise dental antibiotic prescribing [9]. Facilitating shared decision has proved a successful way to reduce antibiotic prescribing in primary medical care [10], but is untested in dentistry. To inform the development of new antibiotic stewardship approaches for primary dental care, several studies have identified potentially modifiable factors which influence the decision whether to prescribe antibiotics which could be suitable for inclusion in a behaviour change intervention [9,11]. Whilst many of the thirty-one factors identified were common across both medical and dental settings, some were previously unreported in any primary healthcare setting. Examples included: the clinician’s goal to fix the problem during the urgent appointment (so as to prevent it recurring) rather than just providing symptomatic (e.g., pain) relief, and clinicians’ beliefs about whether providing a procedure (such as draining an infection) was possible during an urgent appointment.

In 2016, a dental antimicrobial stewardship toolkit was introduced in England to provide free, online access to guidelines, information and training about dental antibiotic prescribing and resistance [12]. Large gaps in the toolkit have been identified, however, between the thirty-one factors influencing antibiotic prescribing by dentists and the relatively few factors (mainly clinician knowledge) addressed in the toolkit (through clinician guidelines, education and self-audit) [13]. Significant potential exists, therefore, to design a new dental antibiotic stewardship tool to complement those within the existing toolkit, especially in relation to clinician beliefs, professional identity and influence by other people.

The purpose of this paper is to report the development of an evidence-based, behaviour theory-informed, shared decision-making tool to optimise antibiotic prescribing by dentists, for adults with acute dental pain or infection, during urgent dental appointments, initially in England. In line with the ethos of shared decision-making (where equal partnerships and patient empowerment are key), a co-development approach with dentist, patients and other stakeholders was chosen. If shown to be successful at reducing dental antibiotic prescribing, this tool will be translated into other dental contexts worldwide to contribute towards global efforts to tackle antimicrobial resistance.

## 2. Results

### 2.1. Stage 1—Understanding the Behaviour/Prioritising Factors

Dentists, patients and the other stakeholders reached a consensus on prioritisation of nine factors (from thirty-one factors identified in a published ethnographic study [11]) for inclusion in this new dental antibiotic stewardship tool: ‘antibiotic beliefs’, ‘competing demands’, ‘fix the problem’, ‘patient influence’, ‘patient management’, ‘peers and colleagues’, ‘planning and consent’, ‘procedure possible’ and ‘professional role’. Of these, seven had also been identified previously in a systematic review of factors associated with dentists’ decision whether to prescribe antibiotics for adults with acute dental conditions [9].

To underpin intervention development, the first stakeholder meeting began the process of prioritising the factors associated with the decision whether to prescribe dental antibiotics. Having also reviewed antibiotic stewardship interventions developed for use in the primary medical care context, the stakeholders recommended translation of two key aspects for the new dental antibiotic stewardship tool:

(1)Engaging patient in (rather than just giving them a leaflet or telling them the treatment decision) during urgent dental appointments; and(2)The use of diagrams on a leaflet (as per the Royal College of General Practitioner’s Urinary Tract Infection self-management leaflet of the Treat Antibiotics Responsibly: Guidance, Education Tools (TARGET) toolkit) to nudge and assist the dentist to explain the diagnosis to the patient [14].

### 2.2. Stage 2—Identification of Behaviour Change Techniques

Based on the mapping of each factor to domains of the Theoretical Domains Framework (TDF) (as per the original publication [11]) and using the Theory and Techniques Tool (TTT) [15], candidate behaviour change techniques (BCTs) were identified. BCTs that were assessed to fit with a shared decision-making approach are presented in Table 1. Identification and assessment of the candidate BCTs are presented in detail within Appendix A.

A worksheet format for the tool was chosen to engage the patient in the decision-making process and included diagrams to explain the diagnosis (as advised by the stakeholders in Stage 1). A draft worksheet (see Appendix A) was developed, incorporating the identified factors and BCTs, (see Table 1) including information about salient health consequences of unnecessary antibiotic prescribing. The draft worksheet was structured to prompt elicitation of patient preferences and values (an essential component of shared decision-making [16]). The presence of the worksheet in urgent dental clinics would add an object to the environment to help deal with competing demands, and at the same time, act to restructure the social environment relating to both patients and colleagues across the whole dental team. Recognisable logos within the worksheet would provide a credible source to support the dental professionals accepting the tool as relevant to their role and identity.

### 2.3. Stage 3—Planning to Deliver the Tool and Acceptability Testing

During the think aloud study, the stakeholders provide advice about the content and structure of the worksheet which informed modification of the draft worksheet (see Table 2).

The think aloud study also demonstrated to the researchers that the folded design (to reveal the panels to the patient in a specific order) was too complex. Printing out the leaflet and folding it in the right way was, therefore, a barrier to its use in the intended way. This informed redesign of the worksheet to a more flexible two sides of A4 which could be printed back-to-back and folded if desired. Following feedback from the dentists and dental nurses, colours for the leaflet were also chosen which appeared equally well when printed in black and white as colour, as not all dental practices have colour printers.

#### 2.3.1. Description of the Worksheet

The worksheet to facilitate shared decision-making consisted of six sections printed ‘back to back’ on a single sheet of paper. Side one consisted of three sections (see Figure 1) and was designed for use by the patient whilst waiting to meet the dentist for their urgent dental care. Side 2 also contained three sections (see Figure 2) and was designed for the dentist to complete with the patient during the appointment, and for the patient to take home afterwards.

Each section of the worksheet was linked to the priority BCTs, as detailed in Table 3 and, additionally, Section 4 contributed to delivery of the UK Government’s 20-year vision for tackling antimicrobial resistance through diagnostic stewardship by nudging dentists to make a diagnosis.

#### 2.3.2. Planning to Deliver the Worksheet Tool as Part of a Wider Intervention

Several of the BCTs prioritised by the stakeholders for intervention development were not encompassed within the worksheet: instruction on how to perform the behaviour, verbal persuasion about capability and goal setting (behaviour). As highlighted by one of the dentist participants in the think aloud study (Section 2.3), the dentists using this tool would require training, for example to learn how to explain the risks of antibiotic use, such as ‘antibiotic-related colitis’.

An online, motivational training package for dentists to accompany use of the tool is being developed, therefore, as part of the shared decision-making, dental antibiotic stewardship intervention. As well as introducing the dentists to using the worksheet, it will address instruction and persuasion relating to dentists’ skills in ‘patient management’ (including the processes of diagnosis, treatment planning and consent) and their beliefs about their capabilities for diagnosing, treatment planning, gaining consent and providing dental procedures during urgent appointments. It will also set the goal of urgent dental appointments to be about ‘fixing the acute dental problem’ with definitive treatment wherever possible (in line with the NHS commissioning standard for urgent dental care [17]), rather than just aiming to provide symptomatic relief of pain or infection. This online training is being developed in collaboration with Health Education England (the UK Government’s public body which provides education and training to the health workforce) and will be delivered by a source seen a credible by NHS dentists providing urgent dental care in England, such as an academic institution or professional body.

### 2.4. Logic Model for the Shared Decision-Making, Dental Antibiotic Stewardship Intervention

A logic model for the shared decision-making, dental antibiotic stewardship intervention (incorporate the worksheet, together with an accompanying online, motivational training package) is presented in Figure 3. The logic model demonstrates how the inputs, activities and participants are intended to deliver impact.

## 3. Discussion

A shared decision-making tool, which is tailored to urgent NHS dentistry in England and acceptable to patients and dentists in this context, has been developed to optimise antibiotic prescribing by dentists for adults with acute dental pain or infection. A stakeholder group of patients and members of the dental team co-produced the worksheet and focused content on the behavioural influences which they judged as the highest priority to be tackled. Further work to develop an accompanying online motivational training package is underway. Evaluation of the intervention (both worksheet and training) during urgent dental appointments will follow.

Using an evidence-based system for intervention development was an important strength of this study. The behaviour change wheel (BCW) provided the theoretical framework [18,19], with the Theory & Techniques Tool (TTT) proving a useful instrument to support co-development of the worksheet, and streamlining the conventional BCW process by providing a direct link between BCTs and TDF domains [20]. In doing so, it removed the step to ‘identify intervention functions or policy categories’ [18] and enabled focus on just the BCTs which were potentially capable of delivering the desired behaviour change [15].

For development of this shared decision-making intervention (which is about levelling the playing field between patients and clinicians), the researchers viewed sharing responsibility and power between stakeholders and researchers as essential to the ethos of the project. Whilst patient and public involvement and engagement in research are now an expected component of research activity in the UK [21], the researchers wanted to go beyond this requirement, by including stakeholders in all aspects of this project. Stakeholders were first involved as members of the steering group which shaped and oversaw delivery of the research programme from its earliest stages. This included supporting a systematic review study [9] which was undertaken whilst also applying for initial funding of the ethnographic study to understand the factors influencing the decision whether to prescribe antibiotics [11]. Involvement continued and expanded through the stakeholder groups to prioritise the factors for intervention development and participation in the think aloud study reported in this paper. Stakeholder involvement has evolved over time and, whilst some of the original stakeholders have now ceased involvement, new stakeholders have joined as we work towards full-scale evaluation of this intervention.

Co-production has been defined as ‘collaboration in governance, priority-setting, conducting research and/or knowledge translation’ [22]. Tension among the co-production team is a recognised challenge of co-production [20,23]. Attention to power imbalances, difficult discussions about research rigour versus research relevance, and constant monitoring are advocated to help mitigate this risk [24,25]. Through an established relationships with the lead expert by experience (who sat also on the project’s external steering group), use of facilitators experienced in community engagement and development to lead the two workshop sessions, and keeping the project tightly focused on its ambition (and therefore timescales), these challenges were managed and avoided during this study.

Think aloud studies provide a constructive way to show the acceptability of an intervention and have been hypothesised to improve the uptake and adherence thus leading to a greater change in behaviour when implemented in healthcare [26]. Using a think aloud approach is a further strength for this study as it provided feedback from end users (who live and work across England from the southwest to northeast) which enabled fine tuning of the worksheet’s content and presentation for relevance across the NHS England regions.

High rates of antibiotic overprescribing (not in accordance with guidelines) are known to exist across NHS medical and dental care [26,27,28]. Promulgating clinical practice guidelines is known to have limited effect on changing clinician behaviour [29] and bundles of interventions to optimise antibiotic prescribing across healthcare have been shown to be beneficial [30]. Interventions that facilitate shared decision-making between patients and clinicians are included in bundles for use in primary medical care [31], but no such tools have been reported in primary dental care [32]. While our shared decision-making tool is designed to change the prescribing behaviour of dentists, it is recognised that it will also act on the behaviour of patients [33] and other dental team members (e.g., receptionists [34]). By reducing the expectation that antibiotics will be prescribed, the intervention will also address the ‘peers and colleagues’ factor which is reported to influence dentists’ decision whether to prescribe antibiotics [11].

In addition to the 31 dentist-factors associated with decision-making during urgent dental appointments, the ethnographic study feeding into this study also identified 19 patient-factors [11]. The strength of some patient’s desire for antibiotics and their beliefs about the appropriateness of antibiotics for acute dental pain were highlighted [11]. The extent to which patients avoid seeking dental care for dental problems (72% did not consult a dentist) has recently been shown in a study of health-seeking behaviour across healthcare in England [35]. The widespread but mistaken belief that antibiotics are necessary for treating toothache, and an appropriate way to avoid dental treatment, has been demonstrated in a view of social media posts of Twitter and Facebook [33].

Some patients in the ethnographic study exhibited well-developed negotiation/communication skills [11], and bargaining for antibiotics has also been identified in studies undertaken in medical contexts [36,37]. Empowering patients by supporting communication between patient and clinicians has been identified in NHS primary medical care as another potentially important way of reducing overprescribing of antibiotics [38]. By facilitating shared decision-making, it is anticipated that the worksheet developed will also help improve dentist-patient communication, which has been identified as a priority for oral and dental research in England [39].

Other patients in the ethnographic study reported strongly emotional feelings about dental treatment (such as anxiety or phobia) [11]. Studies of dental anxiety have shown that providing the dentist with information of a patient’s heightened anxiety prior to treatment, and involving the patient in this, reduced the patient’s dental anxiety [40]. Anxiety was also identified in the social media review (of antibiotics and toothache) as a driver of antibiotic-seeking behaviour by dental patients [33]. For this reason, the worksheet includes an anxiety scale to facilitate that shared understanding about the patient’s level of anxiety.

Antibiotics do not work for dental pain which is caused by an inflammatory process (such as pulpitis) [41]. Treatment of non-vital, abscessed teeth with antibiotics can only provide temporary relief (at best) and symptoms inevitably recur [42]. During the COVID-19 pandemic, access to dental procedures was restricted in many countries and remote management with advice, analgesics and antibiotics was encouraged [43,44,45]. A UK Parliament report highlighted the impact of this approach which resulted in suboptimal outcomes for patients during the pandemic: ‘patients have been remotely prescribed with antibiotics for their dental problems but have returned with pain or further swelling as the cause of their dental problem has not been properly addressed’ [7]. It is anticipated that our shared decision-making tool will reduce antibiotic prescribing by increasing the number of dental procedures provided, in accordance with guidance.

The ethnographic study which underpins this intervention development also suggested that it takes longer to deliver dental procedures than antibiotics during urgent dental appointments [11]. A scenario-based questionnaire study has also shown that appointments shorter than 20 min are a risk factor for inappropriate antibiotic prescribing (not in accordance with guidelines) [46]. Furthermore, the National Institute for Health and Care Excellence guideline on shared decision-making places emphasises on providing enough time for people to make the decision that’s right for them [16]. Further research is required, therefore, to test whether shared decision-making and guideline congruent care can be effectively delivered during the 15-min urgent dental appointments commissioned by the NHS in England [17], or whether longer appointments are required.

A limitation of this study is that the shared decision-making, dental antibiotic stewardship intervention resulting from this study focused on just 9 of the 31 factors identified as influences on dentists’ decisions whether to prescribe antibiotics. A number of the factors not addressed by this intervention link to the wider environment context for urgent NHS dental care provided in England. These healthcare system factors include the access to (availability of) routine and specialist NHS dental services required to complete definitive treatment started during urgent dental appointments (such as root canal or provision of a denture to replace missing teeth). The lack of routinely collected dental prescribing data also makes it impossible to hold NHS dental contractors accountable for their dental antibiotic prescribing rates or to financially incentivise optimal urgent dental care through the NHS dental contract [13]. The importance of these health service levels factors was highlighted during the COVID-19 pandemic, when restricted access to dental procedures for treating acute dental pain and infection resulted in a dramatic 25% increase in dental antibiotic prescribing in England [43] compared to reductions in antibiotic use across all other parts of the NHS [47]. For such a health services-wide approach to be delivered for NHS dentistry in England, a complexity of changes (both legislative and technological) would be required. Further research to develop health services approaches for NHS dentistry (such as Quality Premium payments similar to those introduced to primary medical care in 2015 [48]) should be undertaken in preparation for the introduction of systems and processes to facilitate the routine collection of high-quality dental prescribing data, such as via an electronic prescribing system.

If shown to have the desired impact, translation of this intervention into other healthcare contexts would be straightforward. With seven of the nine factors targeted by this intervention having been identified in an international systematic review of dentists’ decision-making about whether to prescribe antibiotics to adults with acute dental pain or infection (and the other two not previous reported on in dental studies), it is likely that translation between dental contexts will be successful. Testing of the intervention in urgent dental care in NHS England and beyond will be needed to assess the efficacy of this tool as an antibiotic stewardship intervention which could contribute to global efforts to tackle antibiotic resistance.

## 4. Materials and Methods

Development of the tool followed the Behaviour Change Wheel (BCW) approach [18], in three stages: (1) Understanding the behaviour and prioritising focus for action; (2) planning to produce the intervention; and (3) planning to deliver the intervention and acceptability testing. BCW encompasses a coherent suite of theories, techniques and tools which combines the plethora of existing behaviour theories and models to facilitate development of behaviour change interventions [18,49]. It includes the Theoretical Domains Framework (TDF), and Theory & Techniques Tool (TTT). TDF provides a comprehensive, theory-informed approach to identify determinants of behaviour and support behaviour change intervention design [50]. The TTT can be used to identify links between TDF domains and behaviour change techniques (BCTs) based on evidence from the literature, expert consensus or triangulation [20].

### 4.1. Stage 1—Understanding the Behaviour/Prioritising Factors

Understanding dentists’ antibiotic prescribing behaviour was based on 31 factors which had been identified in a previous ethnographic study about influences on treatment decisions (including but not limited to antibiotic prescribing) during urgent NHS dental appointments in England [11]. A stakeholder group of experts by experience of urgent dental care (i.e., patients), general dental practitioners (GDPs), dental nurses, NHS service managers, and healthcare researchers took part in two workshops to understand and prioritise factors for inclusion in one or more interventions. In total, 19 stakeholders were involved, including 4 experts by experience, 4 GDPs, 2 dental nurses, 4 managers, and 5 researchers). At the first workshop, three groups of 5 or 6 stakeholders worked with flashcards to become acquainted with each of the 31 factors. The groups were also asked to identify elements of existing antimicrobial stewardship tools [14,35,46,47,51] which they would like to see translated into the new intervention for dentistry. A scribe from www.liveillustration.co.uk made a graphic (cartoon) record of the meeting, which acted to both collate ideas and stimulate discussion among the whole group about the factors and initial feelings about relative importance of each factor. The second stakeholder workshop was organised by a theatre arts charity (www.theatreofdebate) and was designed to broaden and deepen understanding and insight among the group about the factors before then prioritising them as targets for intervention development. To achieve consensus among the group about the priorities, eight pairs of stakeholders ranked their top three priorities, which they then shared and discussed among the whole group. Finally, through discussion, the group decided on the factors which they identified as priorities to include in interventions.

### 4.2. Stage 2—Identification of Behaviour Change Techniques

Each factor had been mapped to the TDF in previous studies [9,11]. The TTT was used to identify candidate BCTs directly from the TDF domains. The suitability of each candidate BCT for inclusion in the final intervention was assessed using the Affordability, Practicability, Effectiveness, Acceptability, Side effects/Safety, Equality (APEASE) criteria, as per BCW guidance [18] (see Appendix A).

A draft worksheet was developed incorporating these BCTs and with reference to frameworks for shared decision-making [16] and advice from the stakeholder groups). The results were summarised in a logic model for the tool to facilitate shared decision-making between patients and dentists in order, ultimately, to contribute to the UK’s delivery of the WHO global action plan on antimicrobial resistance [1].

### 4.3. Stage 3—Planning to Deliver the Tool and Acceptability Testing

A draft worksheet was developed by the principal researcher (WT), incorporating the BCTs and other elements identified during the stakeholder workshops. A think aloud study with key stakeholders (patients, dentists and dental nurses) was undertaken through interviews to check the content, to review preferences for the mode of delivery (face-to-face with individuals or at distance with people accessing the intervention digitally) [18] and to fine tune the presentation of the material. A copy of the participant information sheet for the think aloud study, including questions used to collect feedback during the interviews, is provided as Appendix A. Following incorporation of the results of the think aloud study into a revised draft of the worksheet, the new draft was sent for Crystal-Mark approval (by www.plainenglish.co.uk) to confirm the clarity of the worksheet.

Ethical approval for the study was gained from the University of Leeds Dental Research Ethics Committee (DREC ref: 101218/WT/267 dated 18 December 2018).

## 5. Conclusions

A shared decision-making tool, comprising a worksheet with multiple behaviour change techniques built into the text, image content, and mode of delivery, has been produced which aims to reduce antibiotic prescribing for adult patients with toothache or infection during urgent NHS dental appointments in England. The next step will be to evaluate it at point of care and to translate it into other dental contexts so that dentistry around the world can contribute to international efforts to tackle antibiotic resistance.

## Figures and Tables

**Figure 1 antibiotics-10-01345-f001:**
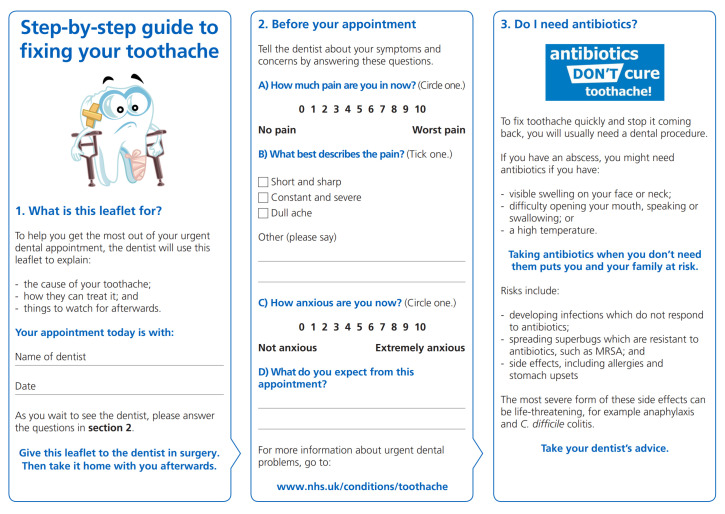
Side one of the shared decision-making worksheet, coproduced to optimise dental antibiotic prescribing during urgent dental appointments. Reproduced with permission of University of Leeds.

**Figure 2 antibiotics-10-01345-f002:**
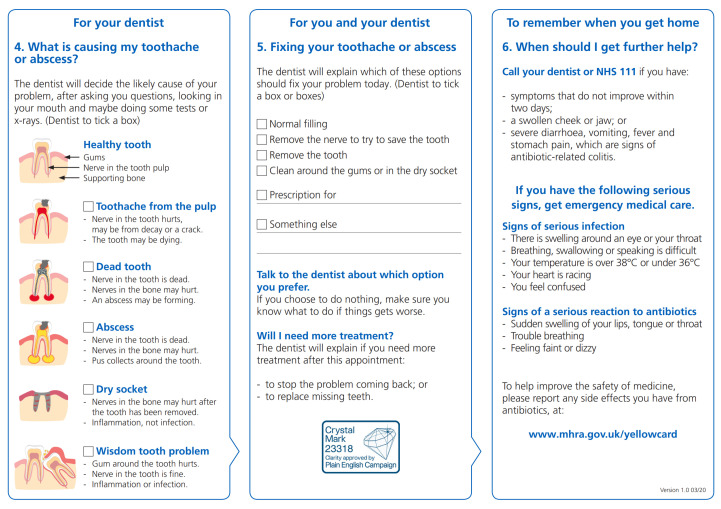
Side two of the shared decision-making worksheet coproduced to optimise dental antibiotic prescribing during urgent dental appointments. Reproduced with permission of University of Leeds.

**Figure 3 antibiotics-10-01345-f003:**
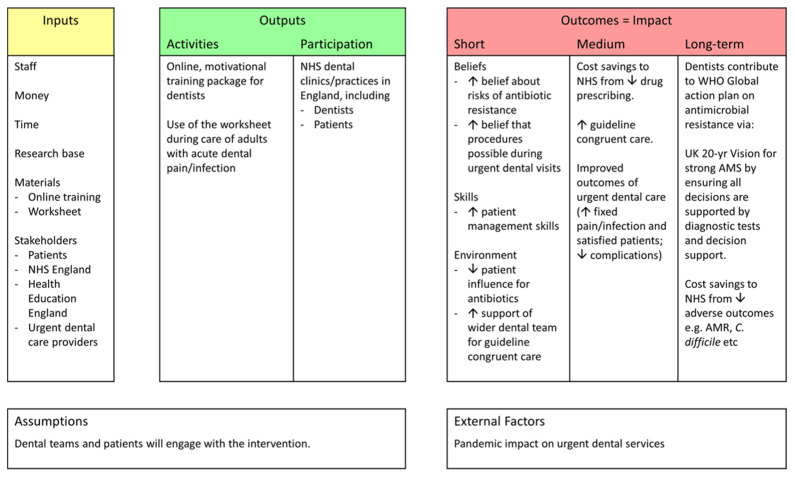
Logic model for the shared decision-making, dental antibiotic stewardship intervention (including the worksheet tool and accompanying online, motivational training package).

**Table 1 antibiotics-10-01345-t001:** Factors that affect decision-making by dentists during urgent dental appointments prioritised for intervention development, mapped to the Theoretical Domains Framework (TDF) and with the selected Behaviour Change Techniques (BCTs).

Priority Factors	TDF	BCT
Antibiotic beliefs	Beliefs about consequences	Information about health consequences Salience of health consequences
Competing demands	Environmental context and resources	Prompts and cues Restructuring the social environment Adding objects to the environment
Fix the problem	Goals	Goal setting (behaviour)
Patient influence	Social influences	Restructuring the social environment
Patient management	Skills	Instructions on how to perform the behaviour
Peers and colleagues	Social influences	Restructuring the social environment Structuring the physical environment
Planning and consent	Beliefs about capabilities	Verbal persuasion about capability
Procedure possible	Beliefs about capabilities	Verbal persuasion about capability
Professional role	Professional role and identity	Credible source

**Table 2 antibiotics-10-01345-t002:** Details of feedback from the stakeholders and resulting modifications made to the draft worksheet.

Worksheet Section/Issue	Feedback from Stakeholders	Resulting Modification
Title and ‘What is this leaflet for? section	Suggested title: ‘Step-by-step guide to fixing your toothache’. Add the dentist’s name.	Title changed accordingly and a space for the dentist’s name (to be written by the clinic’s reception team) was added.
‘Notes’ section	More structure required and located earlier in the worksheet. ‘*Provide more structure to this section to get me to think about my problem. And then ask me if I think I might need antibiotics!*’ (Patient participant) A dentist participant suggested asking patients about their anxiety which can be a problem in urgent dental appointments.	New section added: ‘Before your appointment’ and included visual analogue scales for pain and anxiety, plus free text for patient’s expectations.
‘Do I need antibiotics?’ section	Essential section. Advised locating where it could be read by patients whilst waiting to see the dentist without being the primary focus of the worksheet.	Relocated after the ‘Before your appointment’ section.
‘What is causing my dental problem?’ section	Diagrams essential to help explain the cause of symptoms to patients, but could be clearer: ‘…*don’t need to be a double tooth. Just include a healthy one at the top*.’ (Patient participant)	The diagrams were simplified in line with the advice.
Issue of credibility	Credible source essential. A dentist participant noted ‘*It needs to look official. Good quality paper. And can you add the NHS logo?*’	Permission was obtained to use the recognisable ‘Antibiotics Don’t Cure Toothache’ logo from the UK dental antimicrobial stewardship toolkit, and the Crystal Mark added credibility to the worksheet.
Issue of format	Concerns about a digital version included digital exclusion. Concerns about a paper-based version included colour printing, as some dental practices only have black and white printers.	A paper-based format with colours which are clear when printed in either colour or black and white.
Issue of dentist’s ability	Training need identified: ‘…*training to teach dentists how to explain things like antibiotic-related colitis*.’ (Dentist participant)	During implementation, the on-line training package to accompany the worksheet would cover this skill.

**Table 3 antibiotics-10-01345-t003:** Details of the content and behaviour change techniques (BCTs) within each section of the shared decision-making, dental antibiotic stewardship tool.

Section	Content	BCT
1. What is this leaflet for?	Reception completes Section 1 and hands the worksheet to the patient, thus engaging them in optimising urgent dental care.	Restructuring the social environment Adding objects to the environment
2. Before your appointment	Patient completes their pain, anxiety and other information, and hands the worksheet to the dentist, thus prompting the dentist to dedicate time to understanding the patient perspective.	Prompts and cues
3. Do I need antibiotics?	Provides information about the risks of antibiotics. ‘Antibiotics Don’t Cure Toothache’ branding adds credibility	Information about health consequences Salience of health consequences Credible source
4. What is causing my toothache or abscess?	Environment restructured so the patient expects to be told a diagnosis. Diagrams nudging and assisting the dentist to explain the diagnosis.	Restructuring the social environment Prompts and cues
5. Fixing your toothache or abscess	Prompts the dentist to explain treatment options and share decision-making. Empowers the patient. The Crystal Mark adds credibility to the workbook.	Prompts and cues Credible source
6. When should I get further help?	Provides safety netting advice (Information about what to do if the treatment provided fails). Referencing the Medicines and Healthcare products Regulatory Agency adds credibility.	Information about health consequences Credible source

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
