# Peer review of "Co-Developing an Antibiotic Stewardship Tool for Dentistry: Shared Decision-Making for Adults with Toothache or Infection"

_antibiotics, 2021, doi:10.3390/antibiotics10111345_

Round 1

Reviewer 1 Report

Thank you for presenting this interesting research where the framework for antibiotic stewardship was tested with a different approach. 

Author Response

Thank you for your positive review.

Reviewer 2 Report

The authors should be congratulated on their efforts to create a stewardship intervention which seeks to address the very root causes of inappropriate antimicrobial use in the context of dentistry, a crucial but somewhat neglected field within stewardship. I appreciate the detailed description of the behavioral methods used, which are remarkable. However, I would caution the authors that they may need to explain some of these concepts so that the average reader interested in dental stewardship but not familiar with these methods will still find the article accessible. This is important to ensure that the methods are capable of being reproduced in another setting. I look forward to seeing further work from the authors describing how this proposed intervention performed in practice. 

Line 66: Could elaborate on what the gaps are, so that I understand why an additional intervention was necessary.

Line 128: This is a long but important section. Consider whether breaking down some of the main points of what the dental and patient participants suggested into a table may help. You could also include the quotes in the table in a separate column. It would then be up to you to decide if you could simplify the text, based on what goes in a table. 

Line 200: Is this table referenced in the text? 

Line 211: I believe this section would make more sense toward the end of the discussion as part of suggestions for future work, rather than in the results. However, if you want to leave it here because it is part of the current intervention but not yet completed, it may be better to replace the conditional mood statements with the indicative. For example, instead of "a training package should form part of the intervention" you could write something like "a training package is currently being developed as part of this intervention."

Line 224: Just curious if this was used as part of the intervention, or if this is strictly for purposes of the manuscript. 

Line 229: You probably don't need to include your motives again here. You explained that well in the introduction and will again in the discussion. 

Figure 3: What does HEE stand for, under stakeholders? 

Line 245: The reader may not be familiar with the COM-B model. 

Line 253: You say shared decision-making is about leveling the field between patients and clinicians, but it doesn't seem that the rest of the paragraph reflects their involvement in your study, though they seemed to be a key component. 

Line 275: This paragraph may not be necessary, or else can be shortened significantly, as it restates elements from the introduction. You could incorporate the new references listed here into your introduction.

Line 306: Julie Szymczak and Jeff Linder among many others have written extensively about this topic and may be references worth including.

Line 310: I am not familiar with the James Lind Alliance.

Line 398: This section was hard to follow. There are many references to outside sources which the average reader is unlikely to have any knowledge of. While you provide the sources and references, it would be better, if possible, to provide a brief description of these. This would include the TDF, the TTT website, the APEASE criteria, possibly TARGET, and COM-B described in the discussion. These are first referenced earlier in the results section but may be more appropriate to explain in the methods. Please check to ensure that there are not also a few grammatical errors in this section.

Line 410: Again, I don't think you need to repeat you motives in the methods or results sections. 

Author Response

Thanks for your comments which have helped to improve the presentation of the paper.

Reviewer 3 Report

In this interesting study authors presented a development of an evidence based antibiotic stewardship tool which is a topic that should be of interest to the readers. The paper is well written. Introduction covered the main background points and led up to the main aim of the study. Authors presented the key elements of the study design and adequately described the used methods. Results section reported all described methods described and the results were interpreted and given the broader context in the discussion section. References seem to be appropriate and up to date.

Paper suffers only from some minor issues:

Quality of Supplementary Figure S1 is too low. It contains text written in a very small font which made parts of it unreadable, zooming does not help as the image quality is too low.

Page 4 line 126 “identify” was written instead of “identity”

Page 8 line 204 There is an extra dot in the subheading 2.3.2.

Page 11 line 386 spacing missing between a words and brackets with reference numbers.

Author Response

Thanks for your comments. Please see our response attached.
